# Comparing online versus laboratory measures of speech perception in older children and adolescents

Tara McAllister[1]*, Jonathan L. Preston[2], Laura Ochs[3], Jennifer Hill[4], Elaine R. Hitchcock[3]

1 Department of Communicative Sciences and Disorders, New York University, New York, New York, United States of America, 2 Department of Communication Sciences and Disorders, Syracuse University, Syracuse, New York, New York, United States of America, 3 Department of Communication Sciences and Disorders, Montclair State University, Montclair, New Jersey, United States of America, 4 Department of Applied Statistics, Social Sciences, and Humanities, New York University, New York, New York, United States of America

* tkm214@nyu.edu

## Abstract

Given the increasing prevalence of online data collection, it is important to know how behavioral data obtained online compare to samples collected in the laboratory. This study compares online and in-person measurement of speech perception in older children and adolescents. Speech perception is important for assessment and treatment planning in speech-language pathology; we focus on the American English /ɹ/ sound because of its frequency as a clinical target. Two speech perception tasks were adapted for web presentation using Gorilla: identification of items along a synthetic continuum from *rake* to *wake*, and category goodness judgment of English /ɹ/ sounds in words produced by various talkers with and without speech sound disorder. Fifty typical children aged 9–15 completed these tasks online using a standard headset. These data were compared to a previous sample of 98 typical children aged 9–15 who completed the same tasks in the lab setting. For the identification task, participants exhibited smaller boundary widths (suggestive of more acute perception) in the in-person setting relative to the online setting. For the category goodness judgment task, there was no statistically significant effect of modality. The correlation between scores on the two tasks was significant in the online setting but not in the in-person setting, but the difference in correlation strength was not statistically significant. Overall, our findings agree with previous research in suggesting that online and in-person data collection do not yield identical results, but the two contexts tend to support the same broad conclusions. In addition, these results suggest that online data collection can make it easier for researchers connect with a more representative sample of participants.

## Introduction

Clinical research has identified differences in speech perception between individuals with typical speech and language and individuals who present with speech sound disorder (SSD) [1].

**Data Availability Statement:** De-identified data and code to reproduce the analyses reported in this paper are freely available on the Open Science

Framework at https://osf.io/3rn7m/, DOI 10.17605/OSF.IO/3RN7M.

**Funding:** TM - R01DC017476, National Institute on Deafness and Other Communication Disorders, https://www.nidcd.nih.gov. The funders had no role in study design, data collection and analysis, decision to publish, or preparation of the manuscript. ERH - R15DC019775, National Institute on Deafness and Other Communication Disorders, https://www.nidcd.nih.gov. The funders had no role in study design, data collection and analysis, decision to publish, or preparation of the manuscript.

**Competing interests:** The authors have declared that no competing interests exist.

The speech output of individuals with SSD is characterized by deviations (substitutions, disortions, omissions, and/or additions) that exceed developmental expectations and can negatively impact speech intelligibility. The present study is part of a broader program of research aimed at better understanding differences in the sensorimotor control of speech between children with typical speech and with SSD. In particular, this line of research aims to understand the role of speech perception in speech deviations that persist through late childhood or adolescence. Children whose speech deviations persist past roughly nine years of age may be identified as exhibiting residual speech sound disorder, or RSSD [2]. Speech perception deficits are present in many children with SSD/RSSD, but they are not universally present [1, 3, 4], and accurate assessment of speech perception is important for targeted treatment planning. To support future clinical research, this study investigates the properties of speech perception measures administered in an online context for children with typical speech development. In a previous study, we collected a battery of sensory measures from a group of typically developing children aged 9–15 [5] to serve as a point of reference for children with RSSD in the same age range. The COVID-19 pandemic took place while the larger research program was underway, prompting the study team to investigate options for online collection of measures of speech perception. The present study compares the auditory-perceptual measures collected in-person in the previous study [5] against data obtained when the same measures were administered in the online setting to children in the same age range. Our speech perception measures focus on the /ɹ/ sound (as in words such as "red" and "deer") due to its frequency of occurrence as a target for children receiving treatment for SSD [6].

## The importance of auditory targets for speech

According to current neurolinguistic models of speech-motor control such as DIVA [7], HSFC [8], and FACTS [9], humans learn to speak by identifying the auditory-acoustic characteristics associated with a speech sound, then exploring the mapping between movements of the speech structures and perceptual consequences until they find a motor command that maps onto a given perceptual target. These models predict that speakers who represent a given speech sound with a narrower region in sensory space should also be more precise in their phonetic realization of that sound. A number of empirical studies have supported this idea of links between perception and production across individuals [10–15], although findings of dissociation are also reported [16–19]. Stored representations of speech sound targets are thought to have both an auditory component (i.e., what should be heard when the sound is produced) and a somatosensory component (i.e., what it should feel like to produce the sound). However, auditory targets are thought to emerge first in development, since infants hear others speak before they can feel themselves produce speech; they are also believed to play the primary role in controlling the production of vowels and other sonorant sounds articulated with limited contact between speech structures [7]. Thus, in the present study we focus on the auditory domain, although we acknowledge that somatosensory function is an important part of the broader picture of speech-motor control [20].

Previous literature suggests that there is considerable variability in speech perception even within the population considered typical with respect to hearing and speech production [21, 22]. To measure the acuity of speech perception in a fine-grained fashion, researchers often engage listeners in tasks of identification, goodness rating, or discrimination of speech sounds in syllables or words [13, 23, 24]. Stimuli may be synthetically manipulated or may be naturally produced tokens drawn from diverse talkers. Previous research has established that speech perception tasks can tap either a phonemic mode of perception, in which listeners classify stimuli in relation to categories that can be used to signal a meaningful contrast between words,

or an auditory mode, in which listeners respond to low-level phonetic detail [23, 24]. For the purpose of the present study, we focus on the phonemic level of perception, which is often the focus of perceptual assessment in clinical contexts.

In a typical identification task, a continuum is generated between exemplars representing two phonemic categories (e.g., /ɹ/ and /w/). The steps along this continuum are presented repeatedly for perceptual classification, and listeners' responses are fitted to a logistic function to generate a curve representing categorical perception of the contrast in question. This fitted function can be used to identify the boundary between phonetic categories (i.e., the point at which either response category is equally likely), as well as the width of the boundary region where stimuli are perceived to be ambiguous [3, 15, 25]. The width of this boundary region reflects the consistency with which listeners assign the ambiguous stimuli around the boundary into phonemic categories. A narrower boundary region is suggestive of greater consistency and has been interpreted as evidence of more acute perception. In category goodness judgment tasks, listeners are presented with variations on a target sound and are asked to rate or classify each exemplar in terms of its acceptability as an instance of a target category [26–28]. Responses are compared to judgments considered to represent a gold standard (typically collected from typical adult listeners, often with phonetic training), with a higher percentage of agreement judged to represent more acute perception.

The ability to measure auditory-perceptual acuity is important for the clinical management of pediatric SSD. A meta-analysis of 73 studies reported that over 80% found evidence of speech perception deficits in children with SSD relative to children with typical speech [1]. However, group averages can mask considerable within-group heterogeneity; in the meta-analysis, more than half of studies reporting a group difference in perception also found cases of typical auditory-perceptual performance in children with SSD. Similarly, in a review of 11 studies, Cabbage & Hitchcock [4] reported that the majority of studies reported a range of perceptual abilities in school-aged children with RSSD, ranging from minimal impact to severe deficits. Previous research has reported that direct training of perceptual targets is beneficial for individuals with perceptual deficits, but not for individuals with a deficit that affects only production [29–31]. It has additionally been suggested that children whose SSD is accompanied by perceptual deficits are at increased risk for spelling and reading disorders [34] and should be monitored accordingly. Finally, some studies have reported that differences in perceptual acuity can be predictive of response to treatment [3, 32, 33]. There is also evidence that the association between perceptual acuity and treatment response could be moderated by factors such as age and sex. For instance, Cialdella et al. [3] found a significant association between auditory-perceptual acuity and treatment response in female but not male participants in a retrospective analysis of 59 participants aged 9–15 who received treatment for RSSD.

Because the presence or absence of perceptual deficits is relevant for treatment planning, it is important for clinical purposes to be able to accurately characterize auditory-perceptual acuity at the individual level in children with SSD. However, there are few validated instruments to measure speech perception in children [34], particularly for older children and adolescents. This creates a need to establish reference data on speech perception tasks against which the performance of children with SSD or RSSD can be compared. Our work in the previous published study [5] aimed to address this need by obtaining reference data for multiple measures of /ɹ/ perception in older children and adolescents. However, it cannot be presumed that the same perceptual measures would yield equivalent values when administered in a remote, online context. This represents an important limitation in light of the increasing frequency with which speech pathology assessment and treatment services are delivered online. The present study sought to explicitly test whether measurement of speech perception for typical

children in the online setting would generate values comparable to those observed in a previously conducted study involving in-person assessment.

## Online studies of speech perception

The literature reviewed above demonstrates the importance of experimental studies of speech perception for understanding sensorimotor control of speech. With the increasing prevalence of online data collection in behavioral science, it is important to understand the extent to which valid measures of speech perception can be obtained online. Even before the pandemic, online collection of speech perception data was gaining prominence in linguistics, psychology, neuroscience, and communication disorders [35–41]. Proponents of online data collection praise its efficiency and its potential to reach a more representative range of listeners than lab-based research [40, 42–44]. When studying a small population, such as speakers of a minority language or individuals with a low-prevalence communication disorder, the ability to recruit without geographic limitations becomes particularly valuable. Of course, the events of the COVID-19 pandemic also made it clear that online data collection offers important advantages in terms of flexibility to continue conducting research in times of global crisis.

It is widely agreed that online data are "noisier" than data in the lab setting. The most notable drawback for auditory research is the difficulty of standardizing equipment and playback volume across listeners [40, 45]. In addition, in many online settings, the experimenter is unable to observe the participant and thus has less control over off-task behaviors, distractors, and background noise. However, it may be possible to offset noise in the data by recruiting a larger sample online than would typically be possible in the lab [45, 46]. In addition, new innovations such as tests to confirm that listeners are wearing headphones [47, 48] and auditory reaction timing tools [40] are improving the quality of online data.

Previous empirical research comparing speech perception tasks conducted online versus in the lab has yielded mixed results [49]. A few studies report no difference between online and lab-based listeners: for instance, [50] reported equivalent performance for online and lab-based listeners in a child speech rating task, despite the inclusion of online listeners who did not use headphones. However, a more common finding is that online and lab-based samples do differ in performance, but often not in a way that prevents reproduction of the experimental effect of interest [40, 51, 52]. For instance, Slote & Strand [40] found that online listeners had lower overall accuracy than lab-based participants in a word identification task, but both groups showed similar patterns of accuracy across words. Yu & Lee [53] administered identification and discrimination tasks to assess perceptual compensation for coarticulation in online and lab-based groups of listeners. They found a main effect of modality (online vs in-person) in their models of both tasks, yet this did not prevent them from reproducing the experimental effect of primary interest. Other studies have offered evidence that online data collection may not only add noise to the data, but may also bias response patterns in a specific manner that may obscure experimental effects of interest. For instance, Wolters et al. [54] found different patterns of relative intelligibility across diphone versus HMM synthesized speech for stimuli presented to online versus lab-based listeners. In addition, Yoho et al. [55] found an interaction between listener and talker sex in intelligibility ratings in an in-person but not an online sample of listeners.

Other studies have indicated that experimenters can take actions to improve the quality of data obtained online. In addition to the headphone screening tasks mentioned above, experimenters are encouraged to incorporate checks for attention, either domain-general or domain-specific, or to screen participants based on their reliability across repeated presentation of stimuli [45, 56]. Bianco et al. [57] found lower performance on an adaptive speech in noise task for online than in-lab listener groups, but the addition of a monetary reward for

high performance brought the performance of the online sample up to parity with lab-based listeners. Cooke & Lecumberri [49] used a known listener sample (local university students) for their online tasks involving perception of synthetically manipulated speech stimuli. The online group's accuracy was lower than that of the in-person sample, but parity was achieved after the exclusion of participants who self-reported the use of low-quality headphones. Finally, Shen & Wu [58] found that younger listeners showed a significant difference in performance between online and in-person modalities in a speech recognition in noise task, but no such effect of modality was apparent in older listeners.

One other important consideration for online data collection pertains to the nature of the experimental question of interest. Most published studies of online perception have measured how stimulus manipulations affect perceptual performance at the group level, such as asking how speech recognition differs across different types of synthetic manipulations [40, 49, 51, 52, 54]. It is less common for online studies to focus on the properties of the individual, such as by measuring individual differences in perception. Measurement of individual differences is especially challenging to carry out online because there is no clear way of knowing how much of a participant's performance reflects their intrinsic ability versus their environment or the equipment they are using (background noise, headphone quality, etc.). However, a small number of studies suggest that it may be possible to measure individual characteristics in a stable manner online. Geller et al. [59] administered an online measure of consonant perception in noise on two separate occasions to listeners recruited through the Prolific Academic crowdsourcing platform. They found that average scores across the two administrations had an intraclass correlation of .80 (p < .001). In addition, Yu & Lee [53] measured perceptual compensation for coarticulation in two ways and examined stability across the two measures. The between-task correlation was statistically significant in both online and in-person samples and did not differ significantly between the samples. While these findings of reliability across repeated administrations increase confidence in online measurement of individual-level characteristics, they do not rule out the possibility of exogenous influences that remain stable over time (e.g., a constant level of background noise that affects performance in a consistent way).

## Purpose

The goal of this study was to examine the extent to which measures of speech perception collected online differ from measures obtained in a previous study in the lab setting. To achieve this aim, we compared a previously collected sample of 98 typically developing children and adolescents aged 9–15 tested in the laboratory setting [5] with a new sample of 50 children in the same age range who were recruited and tested online. First, we asked whether mean values of measures of speech perception (boundary width from a phoneme identification task and percent correct from a category goodness judgment task) differed between the online and in-person modality. Second, we investigated whether there were any interactions between modality and participant age or sex, since previous research suggests that impacts of modality could differ over maturation or across speaker subgroups [55, 58]. Lastly, we asked whether there were modality-related differences in the magnitude and direction of associations between identification and category goodness measures. The rationale for this final analysis is that different measures of the same construct (in this case, perception of /ɹ/) are expected to be positively associated, and the degree of association is predicted to be comparable across modalities. Unlike many previous studies of online versus in-person collection of speech perception data, we did not make use of any online crowdsourcing platforms such as Amazon Mechanical Turk or Prolific Academic, since we are interested in a pediatric population and children are not eligible to participate in crowd work.

## Materials and methods

### Eligibility

Participants in both studies were required to be between 9 and 15;11 (years;months) of age at the time of testing and to speak a rhotic variety of English as their dominant or equally dominant language, per parent report. Parent report was also required to indicate no previous diagnosis of hearing loss, major neurobehavioral disorder, or developmental disability. In-person participants were required to pass a pure-tone hearing screening and brief examination of the oral mechanism, while online participants had to pass a commercially available online hearing screening and a brief examination of oral structure and function over video call [60]. All participants completed the Goldman-Fristoe Test of Articulation-3 [61]; they were required to score in the average range for their age and exhibit perceptually accurate production of /ɹ/. As a measure of language function, in-person participants were required to achieve a passing score on the Clinical Evaluation of Language Fundamentals (CELF-5) Screening Test [62]. Online participants were required to score within the average range (no more than 1.3 SD below the mean) on the CELF-5 Recalling Sentences and Formulated Sentences subtests, which can be administered online without loss of validity. Finally, participants in the online study were required to have access to a quiet room with a broadband internet connection. No comparable requirement was imposed in the in-person study. The significance of this and other discrepancies between the online and in-person samples will be addressed in the Discussion section.

### Participants

Institutional review board approval was obtained from the Biomedical Research Alliance of New York (BRANY, protocol number 18-10-393) for the in-person study, and from the Institutional Review Board of Montclair State University for the online study (protocol number FY18-19-1329). Written informed assent was obtained from all participants, as well as written consent from a parent or guardian. Recruitment for the in-person study was achieved through the placement of flyers in community locations, by posts to community mailing lists, and through social media channels. For the online study, all recruitment was accomplished through social media posts and community listserv emails. The recruitment period for the in-person study extended from 09/18/2019-06/30/2022, while recruitment for the online study was carried out from 10/01/2021-03/31/2022.

In the in-person study, a total of 98 participants qualified for the study and completed both visits. Due to experimenter error, 2 participants did not have usable data for the identification task, and 3 participants did not have usable data for the category goodness judgment task; see below for a detailed description of both tasks. In the online study, a total of 50 participants qualified for the study and completed both visits. In the online context, 0 participants lacked data due to experimenter error. Both datasets were checked for outliers (scores > 3 median absolute deviations (MAD) from the group median), but no observations were excluded based on this criterion. The average age of the in-person participants was 12.81 years (standard deviation 1.96); for online participants, the mean age was 12.08 years (standard deviation 1.88). This difference in age was significant in a t-test (t(102.71) = 2.22, $p$ = 0.03). The modality groups also differed in breakdown by sex. The participants who presented for the in-person study had a higher proportion of female participants (59.2% female), whereas online recruitment yielded a slightly more balanced sample of males and females (54% female). In light of these differences between the modality groups, as well as previous research suggesting that factors such as age and sex may matter for perceptual measurement [3, 55], below we will

examine models of perceptual performance that control for age and sex. See online S1 and S2 Tables for individual-level characteristics (age, sex, and GFTA-3 and CELF-5 scores) in each setting, as well as S1 Fig for a histogram of participant age and sex in each setting.

## Stimuli and protocol

Participants in the in-person study were seen by research speech-language-pathologists from one of three sites: Montclair State University, New York University, or Syracuse University. For the online study, participants were in their home setting; they joined a password-protected Zoom call and worked with a clinician from Montclair State University or Syracuse University. Both studies took place over two visits on separate days, each roughly 1–2 hours in duration. Testing sessions were generally longer in the in-person study, which administered measures of somatosensory function (not discussed here) in addition to auditory measures; we return to this point in the Discussion section. The first session administered all tasks to assess eligibility for inclusion, including the hearing screening, oral mechanism screening, GFTA-3 [61], and CELF-5 measures [62]. In the in-person study, participants completed listening tasks while wearing HD 280 PRO Sennheiser headphones (8Hz-25 kHz frequency response) in a sound-shielded booth or quiet room. Participants in the online study were asked to join from a quiet room and to access the internet using a wired (ethernet) connection to their router. Online participants were provided with a standard set of headphones (Plantronics Blackwire C225, 10Hz-10kHz frequency response) for use in listening tasks. Participants were supervised by study personnel during all tasks in both the in-person and online modality. Parents were not involved in study tasks in either modality.

## Identification task

A computerized task, adapted from McAllister Byun & Tiede [15] and used in Ayala et al. [5], was used to assess the consistency with which listeners partitioned a synthetic stimulus continuum into /ɹ/ and /w/ phoneme categories. An artificial continuum from /ɹ/ to /w/ was synthesized from naturally produced tokens of the words *rake* and *wake* elicited in isolation from a 10-year-old female, whose voice provides an approximate match for child participants' speech. STRAIGHT synthesis [63] was used to create a 240-step continuum between these endpoints. Stimuli were selected that span the continuum range in 10 roughly evenly spaced steps, with tighter spacing in the inner two quartiles of the acoustic range to oversample the probable boundary region. Additional details of the continuum generation can be found in Ayala et al. [5]. The identification task was administered using custom PC-based software [64] in the in-person study and using the Gorilla web interface [65] in the online study. In both contexts, participants were initially engaged in a practice phase featuring two stimuli from the extreme ends of the continuum that were not used in experimental trials. They proceeded to the main experimental trials after achieving 5/5 correct on the practice trials or after completing the practice three times, whichever came first. In the main task, participants heard ten continuum steps eight times each (80 trials total) in randomized order. Each trial featured a single token from the continuum presented twice in a row with a 200-ms interval between the repetitions; this double presentation was intended to make the task more robust to momentary fluctuations in attention. Participants responded by clicking on the word they heard (*rake* or *wake*). A brief break was offered after every 20 trials.

An identification function was generated for each child by plotting the proportion of tokens identified as *rake* for each of the ten steps of the main continuum. These ten data points were fitted to a logistic function via maximum likelihood estimation. The location of the phoneme category boundary was defined as the 50th percentile of probability in the fitted logistic

function, and the boundary region was defined as the distance along the continuum between the 25th and the 75th percentile of probability on the fitted function. Consistent with our previous research, we treated the width of this boundary region as our primary measure of auditory-perceptual acuity. A smaller boundary width is indicative of more consistent responding and is interpreted as evidence of more acute perception of the /ɹ/-/w/ contrast. A boundary width of zero means that no more than one continuum step was classified with any degree of inconsistency.

## Category goodness judgment task

Participants' category goodness judgment for /ɹ/ was assessed in a task with 100 naturally produced single words from 52 speakers, including children with and without RSSD affecting /ɹ/, as well as typical adult speakers. In each trial, participants saw a written word containing /ɹ/ and heard a recording of a speaker producing that word; they received the following instructions: "Listen to the word containing /r/. After hearing a word, choose whether the /r/ sound is right or wrong." No training was provided, allowing participants to use their own standard for "right" or "wrong" production. The target words contained a single /ɹ/ sound that could occur as the nucleus of the syllable (n = 28 items), in postvocalic position (n = 22 items), or in onset position (n = 25 items with singleton /ɹ/ and n = 25 items with /ɹ/ in a cluster context). Fifty items had an intended response of 'correct' and 50 had an intended response of 'incorrect,' based on consensus across at least four experienced listeners. Stimuli were standardized with respect to root mean squared amplitude and sampling rate. Stimulus presentation and response recording were carried out using Praat software [66] in the in-person study and using the Gorilla web interface [65] in the online study. No training was provided due to the relatively self-explanatory nature of the task. A brief break was offered after every 25 trials. Performance on this task was assessed by computing the number of responses where the participant's evaluation agreed with the experienced listeners' consensus judgment as a percentage of the total number of trials.

## Analyses

Study data were stored in secure REDCap databases, with double-entry of data to minimize human error. Shapiro-Wilk tests for normality were conducted for each measure (identification, category goodness judgment) and modality group (online, in-person). The null hypothesis of normally distributed data was rejected in all cases except the category goodness judgment measure for the online modality. Therefore, we used nonparametric measures of central tendency and identified observations as outliers if they fell at least three MAD from the group median [67]. To compare average performance across modalities, group means were compared using two non-parametric Wilcoxon rank sum tests, one for each outcome measure (identification boundary width, category goodness percent correct). To examine whether the effect of modality differed based on participant characteristics, each outcome measure was examined in a linear model with predictor variables of modality (online versus in-person), participant sex, and participant age, as well as two-way interactions between study and sex and study and age. Three-way interactions were not included because of known difficulties with their interpretation. In our final research question, we examined pairwise associations between identification and category goodness judgment scores in each modality. Spearman's rho was used due to the non-normal distribution of participant scores. Two-tailed Fisher's Z tests were then used to compare the strength of the correlations in the online and in-person modalities. All data visualization and analyses were carried out in the R programming environment [68]. Complete scripts to reproduce the analyses reported here, along with de-identified raw data, can be found on the Open Science Framework at https://osf.io/3rn7m/.

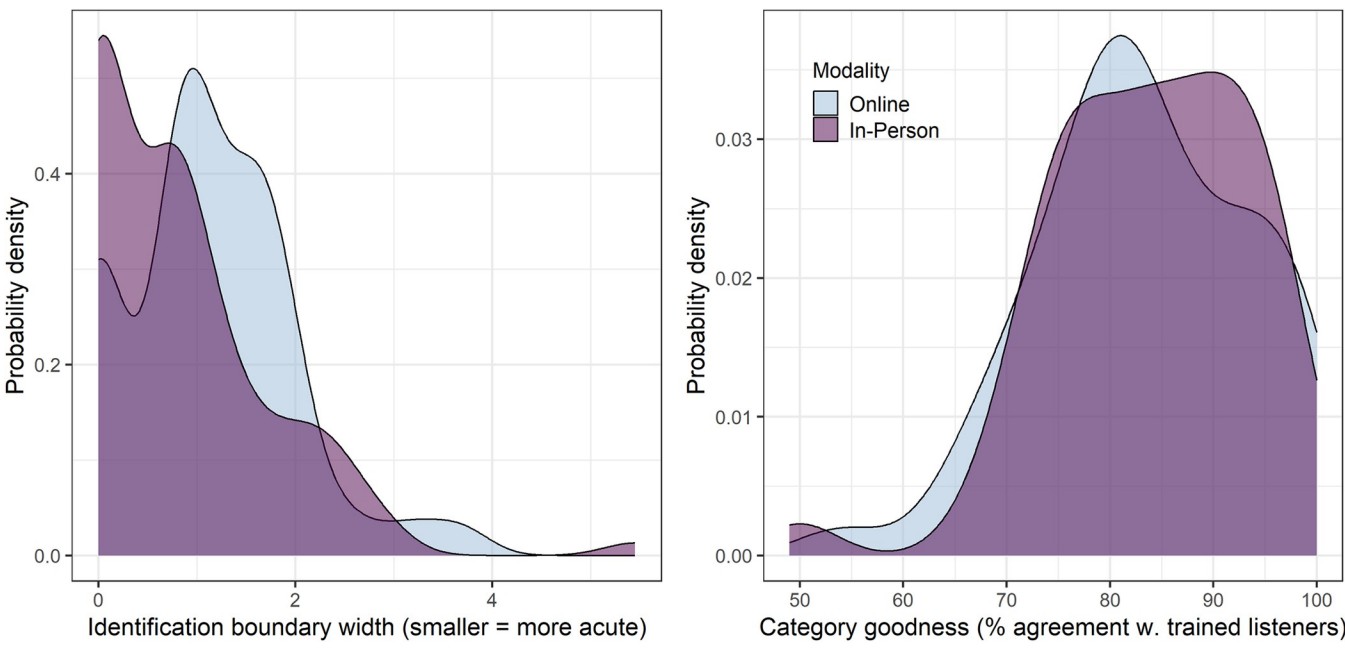

**Fig 1. Distribution of identification and category goodness judgment scores in each modality.**

## Results

Fig 1 shows density estimates of scores on the identification and category goodness judgment task in each modality. As in previous studies using the *rake-wake* identification task adopted here, a ceiling effect was apparent; a boundary width of 0.0 (which means that 0 or 1 stimuli were classified with less than 100% consistency) was calculated for 39 out of 98 participants in the in-person modality (39.8%) and 11 out of 50 participants in the online modality (22%). (Although the distribution of identification scores is truncated at the lower rather than the upper bound, we use the term "ceiling effect" because in this case a low score is indicative of higher performance, i.e. greater auditory acuity.)

We begin our analyses with Wilcoxon rank-sum tests comparing the two modalities with respect to the distribution of scores on the rake-wake identification task and the category goodness judgment task. The means of the modality groups were found to differ significantly with respect to identification boundary width ($W = 1751.5$, $p = 0.0064$). However, there was no statistically significant difference between the modality groups with respect to percent correct on the category goodness judgment task ($W = 2456$, $p = 0.74$).

Because the modality groups differed in age and sex characteristics, it was essential to follow up this first-pass analysis with a model that controlled for these covariates. The linear model for the identification task yielded a significant association between boundary width and experimental modality for female participants of average age (the reference level), such that online data collection was associated with higher/less acute boundary widths than in-person data collection ($\beta = 0.5$, $SE = 0.21$, $p = 0.018$). However, the difference between online and in-person modalities was not significant for male participants of average age ($\beta = 0.2$, $SE = 0.23$, $p = 0.38$). In addition, male participants of average age were found to exhibit higher/less acute scores than females of average age in the in-person modality ($\beta = 0.41$, $SE = 0.18$, $p = 0.023$), but not in the online context ($\beta = 0.28$, $SE = 0.25$, $p = 0.25$). The differential patterning of responses in the online and in-person modalities by sex yielded a significant sex-modality interaction ($\beta = -0.69$, $SE = 0.3$, $p = 0.024$), which can be visualized in Fig 2.

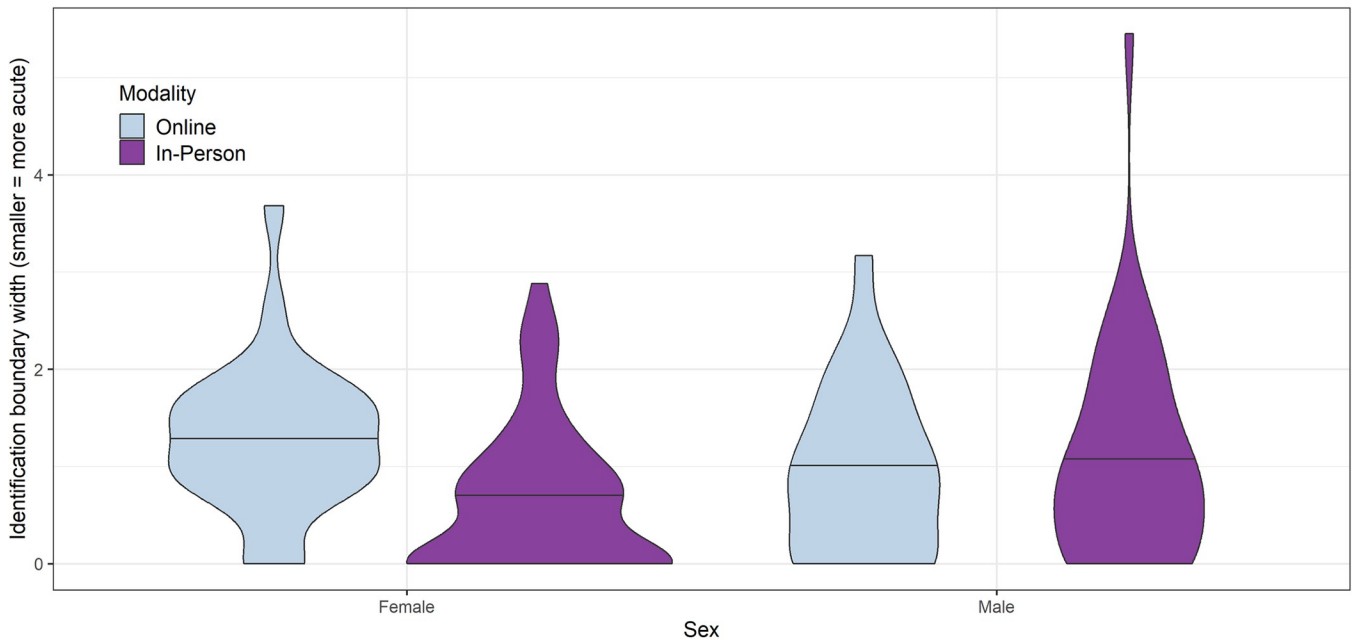

**Fig 2. Interaction of sex and modality for identification boundary width.**

Finally, the model revealed associations between boundary width and participant age and interactions between modality and age. Online participants exhibited lower scores with each year of increasing age ($\beta$ = -0.21, $SE$ = 0.07, $p$ = 0.001); this difference was not significant for those participating in person. A statistically significant interaction between modality and age ($\beta$ = -0.17, $SE$ = 0.08, $p$ = 0.036) indicates that for female participants, the difference between modalities (higher boundary width in the online than the in-person context) is smaller for older participants and larger for younger participants. For male participants, where the average difference between modalities was smaller and in the opposite direction, this interaction means that older participants displayed better performance in the online modality, whereas younger participants displayed a small advantage for in-person data collection. This interaction can be visualized in Fig 3. Complete results for the original and releveled models, measures of goodness of model fit, model diagnostics, and estimated marginal means are provided in the online supplement to this paper (S3 Table).

A linear regression model predicting category goodness scores revealed a nonsignificant trend for females of average age (the reference level) to score lower online than in person ($\beta$ = -2.38, $SE$ = 2.24, $p$ = 0.29). For males, the direction of the association was flipped, but the difference again was not statistically significant. In addition, male participants of average age were found to exhibit lower/less acute scores than females of average age in the in-person modality ($\beta$ = -4.38, $SE$ = 1.96, $p$ = 0.027). In the online modality, female participants had a slightly lower average accuracy than males, but the difference was not statistically significant ($\beta$ = -2.65, $SE$ = 2.66, $p$ = 0.32). This differential patterning was associated with a significant interaction between sex and modality ($\beta$ = 7.03, $SE$ = 3.3, $p$ = 0.035), which can be visualized in Fig 4. There was also a statistically significant association between category goodness scores and age for females and males in the in-person modality ($\beta$ = 1.64, $SE$ = 0.49, $p$ = 0.001). The interaction between age and modality was small and not statistically significant, suggesting that older participants scored higher than younger participants in both modalities. No other effects or interactions were significant at the $p$ = .05 level. Complete model results, measures of model

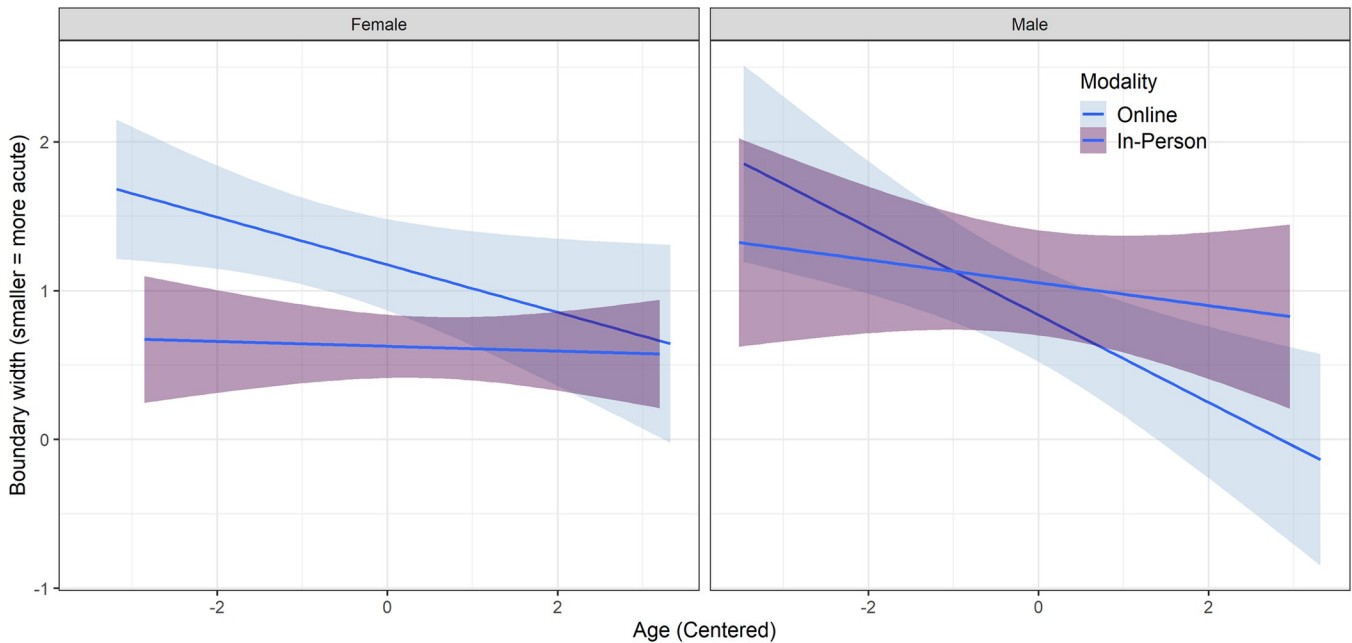

**Fig 3. Interaction of age and modality for identification boundary width, separated by sex.**

fit, model diagnostics, and estimated marginal means are provided in the online supplement to this paper (S4 Table).

The final set of analyses examined pairwise correlations between the two measures of speech perception in both online and in-person contexts. Across the entire dataset (pooling over modalities), there was a small but statistically significant association between

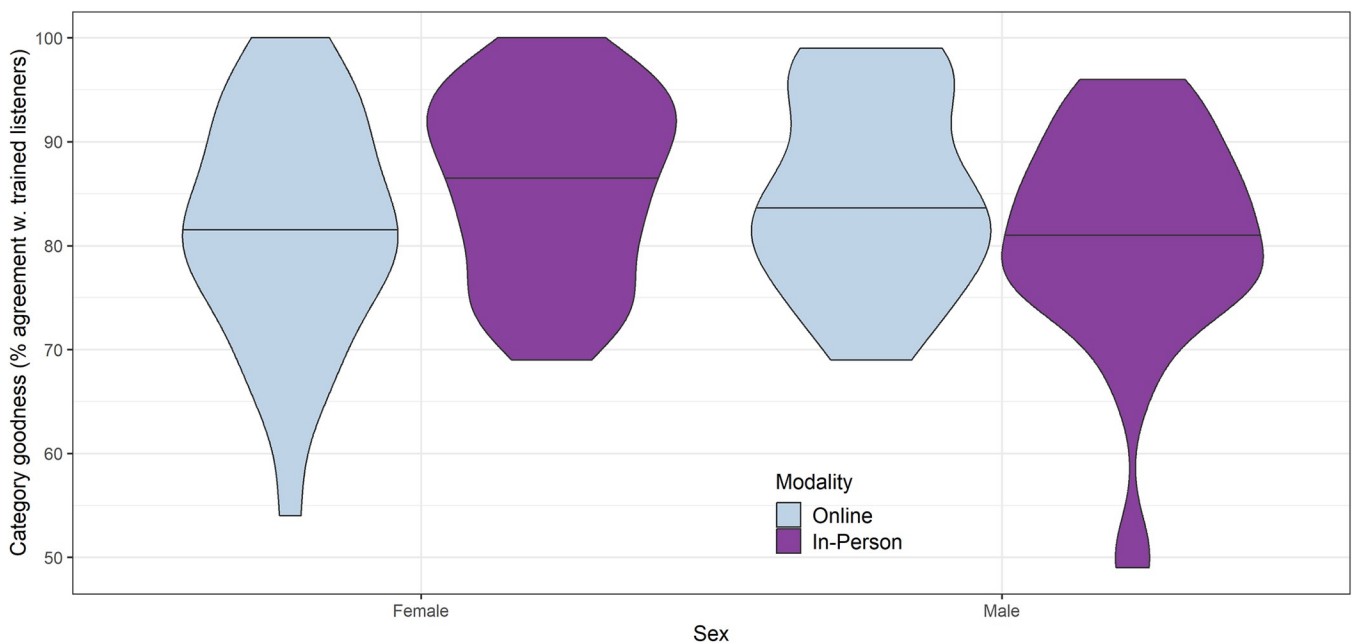

**Fig 4. Interaction of sex and modality for category goodness judgment (percent agreement with trained listeners).**

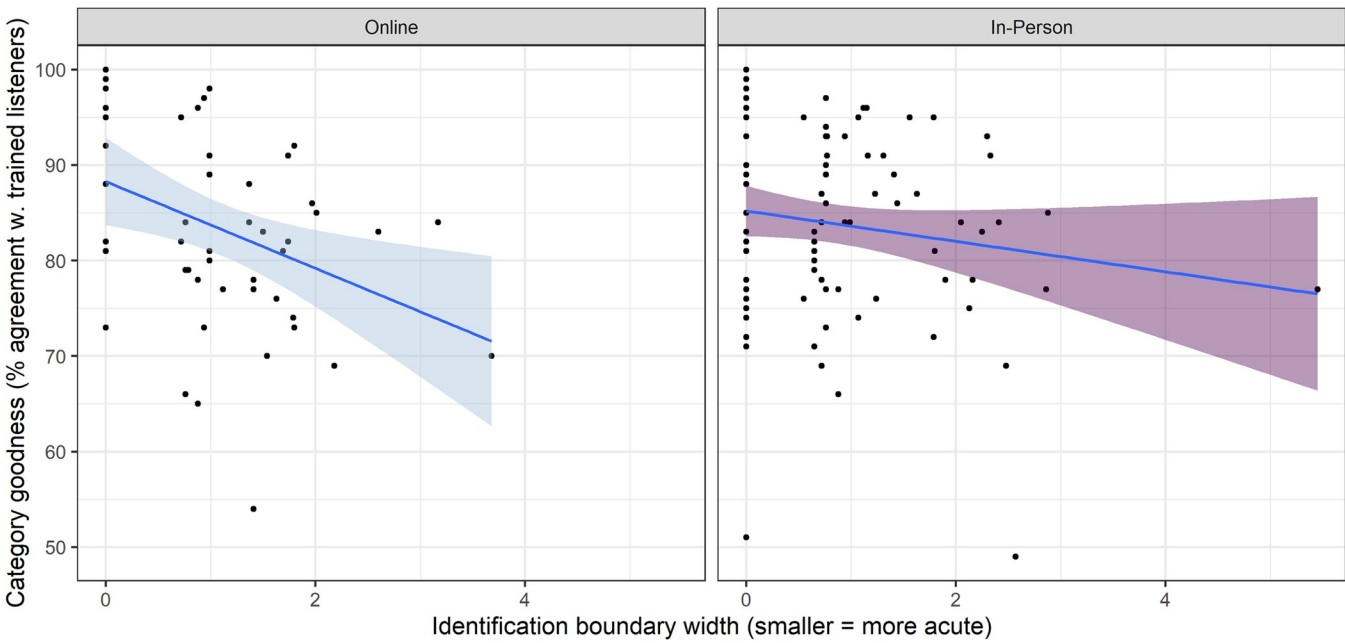

**Fig 5. Pairwise associations between identification and category goodness scores across individuals, online and in-person.**

identification boundary width and category goodness judgment score (Spearman's rho(141) = -0.2, $p$ = 0.02). In the online setting, the association was statistically significant with a slightly greater magnitude (Spearman's rho(48) = -0.34, $p$ = 0.02). In the in-person data, the association was not statistically significant (Spearman's rho(91) = -0.07, $p$ = 0.48). These associations can be visualized in Fig 5. (Some datapoints in Fig 4 present visually as outliers, although they were not flagged as such on the basis of median absolute deviation from the group median. However, the influence of these datapoints is limited by our decision to use Spearman's rho as our measure of association.) However, a two-tailed Fisher's z test indicated that the difference in the strength of association between the two contexts was not statistically significant ($z$ = 1.53, $p$ = 0.13).

## Discussion

This study compared two measures of speech perception administered to 98 children aged 9–15 in the laboratory setting and 50 children in the same age range recruited online. The tasks administered were focused on the American English /ɹ/ sound in light of its high frequency as a clinical target. Tasks included identification of stimuli along a synthetic continuum from *rake* to *wake* and category goodness judgment for /ɹ/ sounds produced by various talkers with and without RSSD affecting /ɹ/. While the modality groups did not differ significantly in performance on the category goodness judgment task, a nonparametric Wilcoxon test revealed that boundary widths on the identification task were significantly smaller (suggestive of more acute perception) for the in-person group than the online participants. Linear regression was used to explore whether performance on either task differed by age and sex, and whether these effects interacted with testing modality. For the identification task, female participants were found to exhibit significantly lower boundary widths (suggestive of higher acuity) in the in-person modality than online; however, male participants showed minimal difference in performance across modalities. For the category goodness judgment task, no statistically significant differences by modality were apparent, although the pattern of positive and

negative trends mimicked what was seen for the identification task. A significant association between age and performance was observed for the identification task in the online setting only, while category goodness judgment scores were associated with age in both modalities. Lastly, scores on the identification and category goodness judgment tasks were significantly correlated in the online setting but not in person. However, the difference in the strength of association across the two modalities was not found to be statistically significant.

Our results also showed differences between male and female participants that varied across modalities. We chose to examine sex and age as predictors based on a small number of previous findings suggesting that biological sex can influence perceptual acuity at a basic level [69] or can interact with perceptual acuity in predicting treatment response [3]. When the results obtained in the in-person setting were reported previously [5], no differences by sex were found for the identification task or the category goodness task. However, in the present study where sex was modeled in interaction with age and modality, the model results suggested that female participants exhibited lower/more acute identification scores and higher category goodness judgment scores than males in the in-person setting. Previous research with adult listeners has suggested that speech perception performance may be more accurate when there is a match in sex between the listener and the model talker [55]. This could potentially account for the difference in performance on the identification task, where stimuli were synthesized using a speech sample from a 10-year-old girl, but stimuli for the category goodness judgment task came from various speakers representing both sexes. In addition, the sex-related differences observed in this study may reflect the influence of differences in the composition of the participant samples recruited online and in person. We consider this possibility in greater detail in our discussion of limitations of the present study.

## Limitations

When interpreting these findings, it is important to consider several limitations of the study. The present study was not designed as a prospective comparison of online and in-person settings. Rather, the lab-based study was planned prior to the COVID-19 pandemic and was executed partly before and partly after the COVID quarantine period. (Because this study reports only results pertaining to speech perception, the introduction of COVID safety measures such as masking are not thought to have a significant impact on our in-person data collected before versus after the beginning of the COVID-19 pandemic.) The online study was inspired by the COVID-19 pandemic. To administer the tasks remotely, it was necessary to use an online platform for experiment administration (Gorilla) instead of the software used in-person (a custom program for the identification task and Praat for the category goodness judgment task). However, the sound stimuli were identical across the online and in-person versions of each task, and the mechanics of task completion were unchanged apart from minor differences in font size and screen color. Therefore, we believe that the difference in software is unlikely to have a large impact on performance across online and in-person modalities. The overall study protocol also differed across modalities, with participants in the in-person study completing additional tasks designed to measure somatosensory acuity; these tasks could not be adapted for the remote modality. The longer study protocol followed by in-person participants could have negatively impacted their ability to sustain attention to the subset of tasks assessing auditory perception. However, we did not observe evidence of diminished attention in the in-person participant group; on the contrary, to the extent that the modality groups differed, the in-person participants tended to exhibit better performance on measures of auditory acuity.

Differences in equipment across modalities could potentially account for our finding that in-person participants tended to exhibit narrower boundary widths (suggestive of higher

auditory acuity) on the identification task. While most online studies of speech perception allow participants to use their own headphones, in this study we aimed to minimize equipment-related variability by mailing a standard set of headphones to all participants. However, there was still a difference in quality between the professional headphones used in the laboratory setting versus the headsets mailed to participants for use in the home setting. In particular, while both devices had over-the-ear headphones, the circumaural closed-back headphones used in the in-person setting can be presumed to have provided more effective sound isolation than the smaller ear cushions featured on the Plantronics Blackwire C225 headset. (The in-person headphones also had a wider frequency response, but we do not consider this to be of material importance for the sonorant stimuli that formed the focus of the present investigation. In research focused on perception of obstruent consonants such as sibilants, the frequency range of the headset may represent a more important consideration.) In addition, it is likely that sources of background noise were better controlled in the lab setting than in participants' homes. Thus, the finding of better performance on the identification task in the in-person modality is likely to be at least partly attributable to these differences in transmission of the acoustic signal.

Interestingly, it can be argued that the slightly lower-quality audio transmission experienced by the online participants actually facilitated the measurement of auditory-perceptual acuity in the specific context of the identification task. It is known from previous research [5] that the identification task is limited by the high percentage of individuals who receive ceiling-level acuity scores (i.e., boundary width of 0.0). Despite this known limitation, the task continues to be used because it has been found to distinguish between children with typical speech and those with RSSD [3] and to predict treatment response among children with RSSD [32, 33]. When administered in the online context, the percentage of respondents who received ceiling-level scores on the identification task dropped from nearly 40% to 22%. It is possible this lower rate of ceiling-level performance contributed to the unexpected finding that scores on the identification and category goodness judgment tasks were significantly correlated in the online setting but not in the in-person context. To achieve a comparable effect in the in-person setting, future research might consider mixing the identification stimuli with a low level of noise to reduce the rate of ceiling-level scores.

Another limitation of this study pertains to differences in the size and composition of the online and in-person groups. The sample size of 98 participants in the in-person study was prospectively planned to provide reference data for sensory skills as a point of comparison for children with RSSD in the same age range. While it would be ideal to obtain the same size of sample in the online modality, our sample size was limited by the funding resources at our disposal. Potentially more important than the difference in group size is the difference in participant characteristics: the group recruited in person had a higher average age and a greater representation of female participants than the group recruited online. This difference in age could have played a role in the finding that average boundary width on the identification task was smaller (suggestive of more acute perception) in the in-person than the online modality. However, the effect of modality remained significant in a linear regression model controlling for participant age and sex. In addition, in a model where the data from online and in-person modalities were reweighted to adjust for differences in sex and age between the two modality groups, there were no changes in the statistical significance of any association, and the magnitudes of the coefficients changed only slightly. (The reweighted model can be found in the code released in conjunction with this manuscript.) Nonetheless, future research investigating the stability of speech perception measures across modalities would benefit from a prospective design in which identical tasks are administered to matched samples of participants online and in-person.

While differences in sex and age between the samples recruited online and in-person represent a limitation of the present study, we consider it worthwhile to reflect further on the observed differences as a possible source of insight into the strengths and drawbacks of each modality. As discussed in more detail in Ayala et al. [5], recruitment for the in-person study was impacted by the COVID-19 pandemic. We had difficulty sampling children under 12 years of age, who were not eligible for vaccination and therefore unlikely to participate in in-person research studies for a significant fraction of the study period. More generally, in the challenging recruitment environment following the COVID-19 lockdowns, we could not be selective in who was included in the in-person sample; we were obligated to enroll all eligible participants in order to approximate our target sample size of 100. As a consequence, the group of participants in the in-person study was not balanced with respect to sex composition (59.2% female), and male and female participants were not equally well-represented at all points in the age range. (See online supplementary S1 Fig for histograms representing the distribution of male and female participants across the age range represented.) In addition, the in-person study had heavy representation of a convenience sample of children of faculty at NYU, whose performance on laboratory tasks might not be fully representative of the general population.

By contrast, recruitment for the online study was not challenging, even in the aftermath of the pandemic. In the online context, it was possible to achieve a more equal balance of male and female participants and ensure that both sexes were comparably represented across the age range. This is consistent with previous research suggesting that online data collection can make it more feasible for researchers to collect data representative of the general population and avoid the biasing effect of local convenience sampling [40]. We also speculated that participants in the online study might be more racially diverse than participants in the in-person study, but participants turned out to have similar racial and ethnic breakdowns in both modalities (*in-person*: 72.7% white, 4.0% Asian, 7.1% Black, 14.1% more than one race, 2% not reported; 6.1% Hispanic or Latino, 86.7% not Hispanic or Latino, 7.1% not reported; *online*: 74% white, 12% Asian, 6% Black, 8% more than one race; 6% Hispanic or Latino, 94% not Hispanic or Latino). However, no explicit measures were put in place to ensure a balanced representation of participants across demographic groups. Our success in achieving better balance with respect to sex and age in the online modality suggests that similar steps could be taken to ensure a demographically representative sample in future online research.

## Conclusions and clinical implications

Although the present study did identify areas of difference between measures of child speech perception collected online and in-person, our results suggest the two modalities yield broadly comparable results. We found that two distinct measures of auditory-perceptual acuity for /ɹ/ were significantly correlated in the online data, and a test of the difference in the strength of association across the two contexts failed to reject the null hypothesis of no difference between modalities. Unexpectedly, the association between the two measures was not significant in the in-person data, but this is potentially attributable to a ceiling effect that was more pronounced in the in-person setting. In addition, models examining perceptual acuity in association with age, sex, and modality suggested that male and female participants differed in average performance in the in-person setting but not online. The observed differences may reflect some influence of the unbalanced sex composition of the in-person sample (nearly 60% female) and the uneven representation of sexes across the included age range. This less-than-optimal breakdown of included participants represents the challenges of recruiting children for in-person studies in the aftermath of the COVID-19 pandemic. In the online setting, the greater ease

of recruitment allowed us to achieve a more balanced breakdown of participants across sex and age categories. In future research, we intend to leverage the greater ease of online recruitment to obtain samples that are more representative with respect to racial and ethnic composition.

In total, the findings of this study are consistent with results from previous research comparing auditory-perceptual measures administered online versus in-person. It is common for such studies to report that results are not perfectly identical between online and in-person contexts. However, previous studies have tended to find that online and in-person samples do not differ in a way that prevents reproduction of the experimental effect of interest; the two contexts tend to support the same broad conclusions [40, 53]. The same is true of the present study: while there were differences in the details of the two contexts, there was no statistically significant difference in the strength of the correlation between the identification and category goodness tasks across modalities. In addition, previous research on the topic of online data collection has highlighted that it can be beneficial in allowing researchers to reach a more broadly representative sample of participants [40, 42–44]. The present findings support the idea that online recruitment can help researchers secure a sample with a more balanced demographic breakdown compared to the distribution of participants who present to the laboratory setting.

Clinically, numerous studies have reported that children with SSD as a group score lower on measures of speech perception than age-matched children with typical speech production [29, 70–73], but there is considerable heterogeneity within the population of children with SSD [1]. Differences in auditory-perceptual involvement could have implications for treatment planning for children with SSD or RSSD. For instance, children with SSD and atypical speech perception may derive greater benefit from an intervention approach that incorporates speech perception training [74] than children who exhibit atypical production in the context of intact perception. As the prevalence of telepractice delivery of speech pathology services continues to grow, it will be increasingly important to provide reference data that will be valid in the online setting. The findings of the present study cannot answer the question of whether it is valid to use reference data collected in the laboratory setting to classify online participants as typical or atypical, because all data reported here were collected from typically developing speakers and no measures of sensitivity or specificity can be computed. Data collection is underway that will allow us to carry out this clinically relevant analysis. In the interim, the present findings provide a caution that online and in-person data cannot automatically be treated as identical, but they also suggest that data collected online are broadly comparable to measures obtained in the lab and have the potential to be more representative.

## Supporting information

**S1 Fig. Histograms representing the distribution of male and female participants across the age range for both in-person and online study settings.**
(TIF)

**S1 Table. Individual-level characteristics of included participants in the in-person setting.**
(DOCX)

**S2 Table. Individual-level characteristics of included participants in the online setting.**
(DOCX)

**S3 Table. Complete results of the linear model examining identification boundary width.** Includes measures of goodness of model fit, model diagnostics, and estimated marginal means.
(DOCX)

**S4 Table. Complete results of the linear model examining category goodness percent correct.** Includes measures of goodness of model fit, model diagnostics, and estimated marginal means.
(DOCX)

## Author Contributions

**Conceptualization:** Tara McAllister, Jonathan L. Preston, Elaine R. Hitchcock.

**Data curation:** Jennifer Hill.

**Funding acquisition:** Tara McAllister, Jonathan L. Preston, Elaine R. Hitchcock.

**Investigation:** Laura Ochs.

**Methodology:** Jennifer Hill.

**Project administration:** Tara McAllister, Jonathan L. Preston, Laura Ochs, Elaine R. Hitchcock.

**Visualization:** Jennifer Hill.

**Writing – original draft:** Tara McAllister, Jonathan L. Preston, Elaine R. Hitchcock.

**Writing – review & editing:** Tara McAllister, Jonathan L. Preston, Laura Ochs, Jennifer Hill, Elaine R. Hitchcock.

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
