## [Decision Letter · Decision Letter 0]

2 Oct 2023

PONE-D-23-23646Comparing online versus laboratory measures of speech perception in older children and adolescentsPLOS ONE

Dear Dr. McAllister,

Thank you for submitting your manuscript to PLOS ONE. After careful consideration, we feel that it has merit but does not fully meet PLOS ONE’s publication criteria as it currently stands. Therefore, we invite you to submit a revised version of the manuscript that addresses the points raised during the review process. Please submit your revised manuscript by Nov 16 2023 11:59PM. If you will need more time than this to complete your revisions, please reply to this message or contact the journal office at plosone@plos.org. Please include the following items when submitting your revised manuscript:A rebuttal letter that responds to each point raised by the academic editor and reviewer(s). You should upload this letter as a separate file labeled 'Response to Reviewers'.A marked-up copy of your manuscript that highlights changes made to the original version. You should upload this as a separate file labeled 'Revised Manuscript with Track Changes'.An unmarked version of your revised paper without tracked changes. You should upload this as a separate file labeled 'Manuscript'.

We look forward to receiving your revised manuscript.

Kind regards,

Simone Sulpizio

Academic Editor

PLOS ONE

Journal Requirements:

Additional Editor Comments:

Both Reviewers have found your work interesting and of relevance, but they also have raised some concerns on the current version of the work. Since both Reviewers provided a clear and detailed review, I'm not going to repeat their points here. In revising the manuscript, I kindly invite you to carefully consider all the points they raised.

Reviewers' comments:

Reviewer's Responses to Questions

**Comments to the Author**

1. Is the manuscript technically sound, and do the data support the conclusions?

Reviewer #1: Partly

Reviewer #2: Yes

2. Has the statistical analysis been performed appropriately and rigorously? 

Reviewer #1: Yes

Reviewer #2: Yes

3. Have the authors made all data underlying the findings in their manuscript fully available?

Reviewer #1: Yes

Reviewer #2: Yes

4. Is the manuscript presented in an intelligible fashion and written in standard English?

Reviewer #1: Yes

Reviewer #2: Yes

5. Review Comments to the Author

Reviewer #1: Summary

In their manuscript entitled “Comparing online versus laboratory measures of speech perception in older children and adolescents”, the authors present data from an online and an in-person behavioural study (the in-person study is also partly described in a previous paper from some of the present authors) concerning speech perception in children and adolescents, suggesting that data collected online are no less valid than data collected in the laboratory.

Both in the online and the in-person study, they implemented a computerized identification task in which English speaking listeners had to report whether they heard the word “rake” or “wake” after listening to resynthesized auditory stimuli (previously recorded from one female speaker) derived from a continuum in which the first phoneme ranged from /ɹ/ to /w/ phoneme categories. Secondly, they implemented a Category Goodness Judgement task in which listeners heard several different words embedding the /ɹ/ phoneme recorded by multiple speakers with and without RSSD (residual speech sound disorder) and judged whether said words were phonetically realized in a correct way.

Results showed that the identification performance differed between the online and the in-person modalities, which authors linked to possible differences in noise level between the two, putatively influencing the percentage of participants with ceiling scores. No differences were reported for the goodness judgment. Additionally, identification performance and the goodness judgment performance correlated in the online task but not in the in-person task; the difference between the two correlation weights was not significant. Lastly, different interaction effects emerged between “Sex”, “Age” and “Modality (online vs in-person)” which authors attempted to link to different distributions of sex and age between the online and the in-person study.

This study shows potential relevance to the field of speech impairments considering the high influence that perception deficits exert on production deficits and treatment outcomes in children & adolescents, but also given the absence of clear norms to evaluate children with RSSD. The authors evidently put great effort in the online data recording which was carried out during the COVID-19 pandemic as i) they managed to record data from children and adolescents, ii) they provided the same headphones to all participants iii) they supervised all data recording sessions via zoom calls.

Before recommending for publication, there are some concerns described below, which authors should address. In particular, the main claim of this work is that data recorded online are no less valid than data recorded in person but it is not clear how the results would precisely support such claim. Secondly, authors indicate several limitations of the study in the discussion section which need to be addressed more clearly by providing references and/or additional analyses.

Major Issues

1) Authors declare that the aim of the study is to test whether measurements of speech perception for typically developing children performed online are comparable to the ones recorded in a laboratory setting multiple times (lines 123,199). However, in the Limitations section they report that “The present study was not designed as a prospective comparison of online and in-person settings.” (line 451-452). It would be appropriate to be clear about the initial aim of the study as soon as possible to help readers in forming a clear picture as well as in understanding how the authors dealt with the continuation of the study despite the multiple problems caused by the COVID-19 pandemic.

2) Line 221: Is parent report a reliable method to check for the presence of developmental disabilities? Can author provide references to support its validity? Are there similar works in which the validity of parent report is compared to the validity of standardized tests or clinical evaluations?

3) Materials and Methods: Details about the psychometric fitting results are missing. Authors should provide the exact fitting formula with all customized parameters (if any, e.g., lapses), plots of the average fitted curves and an appropriate goodness of fit measure. The absence of these details reduces the reproducibility of the study and prevents readers from understanding how well the model fitted to data.

4) Materials and Methods: Is visual inspection a reliable method to determine whether data are normally distributed or not? Author might provide appropriate tests to justify their point. Furthermore, while non-parametric tests are used when comparing means across the online and in-person modalities, linear models are used to evaluate possible effects of different predictors on the dependent variable. A key assumption to implement linear models is normality of model’s residuals. Authors should provide evidence about the normality of residuals to justify the use of linear models. If such condition is not met, they should implement the appropriate analysis method.

5) Discussion: Importantly, author claim that data collected online are no-less valid than data collected in person (lines 541, 542). They back-up this claim by mentioning that identification boundary and category goodness are correlated in the online study but not the in person one (lines 542-545). It is unclear how this finding would justify their claim nor if this specific pattern in the correlation analysis is the only indicator of the validity of data collected online. Relatedly, it is unclear what do the authors exactly mean with the term “validity”. They also suggest that this correlational pattern might be due to different numbers of participants reaching ceiling levels between online and in-person modality. Authors should better clarify the link between results and their interpretations in the discussion section or provide additional analyses/metrics to support their claim (e.g., plotting and/or repeating the analyses by excluding participants showing ceiling performance might be a starting point). This appears as being the central point of the paper, therefore it should be centrally and clearly addressed in the discussion section.

6) Discussion: Results of linear models showed different unexpected patterns concerning several interaction effects of the predictors “Sex”, “Age” and “Modality” which authors mainly attribute to demographic differences in the characteristics of the participants enrolled in the online and the in-person study. It is completely understandable that such differences were not in complete control of researchers considering the COVID-19 pandemic. However, even if statistically significant differences are present in terms of sex and age distributions between online and in-person modalities, considering the sampling bias as the cause of the found interaction effect might be speculative. Authors should attempt at backing-up their claims by providing appropriate evidence from the related literature and/or develop additional analyses to explicitly test if the sampling bias is the source of the unexpected patterns. If possible, authors might try to repeat the analyses by randomly subsampling the in-person database to match the characteristics of the online group in terms of sex and age and verify whether the interaction effects hold. Possible sex differences in speech perception are indeed understudied, thus providing stronger evidence about such patterns might be of relevance to the field.

Minor Issues

7) Lines 42-44: The syntax of the sentence is a bit unclear, please rephrase to clarify. Additionally, authors should clarify the meaning of “speech deviation” to facilitate the understanding of non-expert readers.

8) Line 57: Authors should provide an exemplary English word embedding the /ɹ/ sound as they do later in the manuscript.

9) Lines 61-62: Please clarify the meaning of “the goals driving speech are dynamic regions in auditory-acoustic and somatosensory space”.

10) Line 66: Please provide a definition for “auditory and somatosensory targets” to help non-expert readers.

11) Line 139: Typo, please correct ``noisier" with “noisier”.

12) Line 212: Please provide a reference or an additional explanation backing up the assumption that a stronger correlation between two measures of the same construct in one modality is an index of the robustness of the assessment in that modality.

13) Line 290: Please fix the citation style of (Ortiz, 2018).

14) Line 314-315: Typo, please correct ``choose whether the /ɹ/ sound is right or wrong" with “choose whether the /ɹ/ sound is right or wrong”.

15) Line 332-334: To facilitate the understanding of the rationale of statistical analyses please do not refer to the “first” and “second” research aim but explicitly report again the aims in words.

16) Line 468: Please be more specific about the result instead of using “higher performance”.

17) Results: While authors provide numerical values of beta coefficients when reporting the results of analyses on the linear models (which are usually self-explanatory), also reporting Mean and Std.Dev/Std.Err. (or other appropriate summary descriptive statistics of choice) of the dependent variables might help readers into mentally representing differences between two/more factor levels.

18) Materials and Methods: In the instruction of the category judgement task authors report that participants saw the “choose whether the /ɹ/ sound is right or wrong” sentence on the screen. Were participants already familiar with the meaning of the “/ɹ/” grapheme or were they given additional explanation about IPA notation? Providing details on instructions, especially in these kinds of tasks developed for children and adolescents might help other researchers in developing efficient instructions

19) Materials and Methods: Were parents present during the assessments in online/in-person modality? Please specify.

20) Factor naming is inconsistent throughout the manuscript. Authors sometime use the term “modality” to refer to the modality in which the experiment was carried out (in-person or online), but some other times they use “condition” (line 373, 387,498) or “group” (line 420, 465). Please use consistent factor and levels naming.

21) Figures: Authors provided density plots to represent the found effects in their statistical analyses but they might not be the most appropriate representations, especially when interaction effects are present. The navigation of the results might be facilitated by adding additional figures employing e.g., boxplots/violin plots for categorical variables (e.g., effect of sex) and e.g., points and lines plots when representing the effect of a continuous variable (e.g., age).

Reviewer #2: I’d like to thank the authors for their interesting and well executed study. The methodology is rigorous, and the authors have written with great clarity and precision, which greatly aids in the potential reproduction of the current study. Additionally, I thank the authors for sharing their data and code, which has been presented with detailed metadata and appropriate comments throughout the code to guide the reader through the analysis.

Understanding the impact of research setting (lab vs. online) is of great importance, particularly in light of the COVID-19 pandemic and the increasing shift toward collecting perceptual data remotely. I thank the authors for their valuable contributions in this area.

My primary concern/critique relates to the second research question, specifically regarding the examination of the interaction between age and sex. In its current form, the introduction lacks sufficient justification or rationale for considering the listener sex in the analysis. Moreover, in the discussion section, the authors acknowledge limited evidence supporting the idea that biological sex might influence perceptual acuity. To strengthen the study's introduction and rationale, I’d recommend incorporating this discussion point into the introduction.

As it stands, the investigation into sex differences appears to rely solely on the presence of observed differences in the results. However, without a well-defined rationale for exploring these sex differences, it becomes challenging to discern whether these disparities are indicative of a genuine sex-effect or merely a consequence of dividing the two groups into further subgroups. Clarifying the motivation behind examining sex differences is crucial to the overall validity and interpretation of the study's findings.

For clarity, can the authors give an example of what type of listener response would result in a boundary of 0.0?

The authors stat that the calculated measures were examined for outliers using a criteria of > 3 MAD and reported no outliers were observed. However, there does seem to be some influential bivariate outliers for the correlation analysis (as observed in Figure 4). Could the authors comment on these observations and their decision not to exclude these cases? Did the presence of these outliers inform the decision to use Spearman’s rho?

Minor edits:

There appears to be a word missing in the first paragraph of the introduction, line 41.

In line 364 of the results, when discussing the study-domain effects for male listeners, I’d recommend just saying that the difference between online and in-person modalities was not significant for male participants of average age. This revised sentence would parallel the structure of the previous sentence discussion the study-domain effects for the female listeners.

6. PLOS authors have the option to publish the peer review history of their article (what does this mean?). If published, this will include your full peer review and any attached files.

Reviewer #1: No

Reviewer #2: No

---

## [Author Response · Author response to Decision Letter 0]

30 Nov 2023

PONE-D-23-23646

Comparing online versus laboratory measures of speech perception in older children and adolescents

PLOS ONE

AUTHORS: Confirmed.

AUTHORS: Removed.

Additional Editor Comments:

Both Reviewers have found your work interesting and of relevance, but they also have raised some concerns on the current version of the work. Since both Reviewers provided a clear and detailed review, I'm not going to repeat their points here. In revising the manuscript, I kindly invite you to carefully consider all the points they raised.

Reviewers' comments:

Reviewer's Responses to Questions

Comments to the Author

1. Is the manuscript technically sound, and do the data support the conclusions?

Reviewer #1: Partly

Reviewer #2: Yes

2. Has the statistical analysis been performed appropriately and rigorously?

Reviewer #1: Yes

Reviewer #2: Yes

3. Have the authors made all data underlying the findings in their manuscript fully available?

Reviewer #1: Yes

Reviewer #2: Yes

4. Is the manuscript presented in an intelligible fashion and written in standard English?

Reviewer #1: Yes

Reviewer #2: Yes

5. Review Comments to the Author

Reviewer #1: Summary

In their manuscript entitled “Comparing online versus laboratory measures of speech perception in older children and adolescents”, the authors present data from an online and an in-person behavioural study (the in-person study is also partly described in a previous paper from some of the present authors) concerning speech perception in children and adolescents, suggesting that data collected online are no less valid than data collected in the laboratory.

Both in the online and the in-person study, they implemented a computerized identification task in which English speaking listeners had to report whether they heard the word “rake” or “wake” after listening to resynthesized auditory stimuli (previously recorded from one female speaker) derived from a continuum in which the first phoneme ranged from /ɹ/ to /w/ phoneme categories. Secondly, they implemented a Category Goodness Judgement task in which listeners heard several different words embedding the /ɹ/ phoneme recorded by multiple speakers with and without RSSD (residual speech sound disorder) and judged whether said words were phonetically realized in a correct way.

Results showed that the identification performance differed between the online and the in-person modalities, which authors linked to possible differences in noise level between the two, putatively influencing the percentage of participants with ceiling scores. No differences were reported for the goodness judgment. Additionally, identification performance and the goodness judgment performance correlated in the online task but not in the in-person task; the difference between the two correlation weights was not significant. Lastly, different interaction effects emerged between “Sex”, “Age” and “Modality (online vs in-person)” which authors attempted to link to different distributions of sex and age between the online and the in-person study.

This study shows potential relevance to the field of speech impairments considering the high influence that perception deficits exert on production deficits and treatment outcomes in children & adolescents, but also given the absence of clear norms to evaluate children with RSSD. The authors evidently put great effort in the online data recording which was carried out during the COVID-19 pandemic as i) they managed to record data from children and adolescents, ii) they provided the same headphones to all participants iii) they supervised all data recording sessions via zoom calls.

Before recommending for publication, there are some concerns described below, which authors should address. In particular, the main claim of this work is that data recorded online are no less valid than data recorded in person but it is not clear how the results would precisely support such claim. Secondly, authors indicate several limitations of the study in the discussion section which need to be addressed more clearly by providing references and/or additional analyses.

AUTHORS: We appreciate the reviewer’s thoughtful consideration of our work and suggestions to improve it.

Major Issues

1) Authors declare that the aim of the study is to test whether measurements of speech perception for typically developing children performed online are comparable to the ones recorded in a laboratory setting multiple times (lines 123,199). However, in the Limitations section they report that “The present study was not designed as a prospective comparison of online and in-person settings.” (line 451-452). It would be appropriate to be clear about the initial aim of the study as soon as possible to help readers in forming a clear picture as well as in understanding how the authors dealt with the continuation of the study despite the multiple problems caused by the COVID-19 pandemic.

AUTHORS: We have attempted to be clear throughout the paper that this involves a comparison of data collected during COVID-19 versus previously corrected data (see, e.g., lines 50-56 in the revised/track changes manuscript). Thank you for identifying two points where we did not state this clearly. We have adjusted the wording accordingly. (Lines 123-126 revised to include “in a previously conducted study”; lines 202-203 (previously line 199), “The goal of this study was to examine the extent to which measures of speech perception collected online differ from measures obtained in a previous study in the lab setting.”

2) Line 221: Is parent report a reliable method to check for the presence of developmental disabilities? Can author provide references to support its validity? Are there similar works in which the validity of parent report is compared to the validity of standardized tests or clinical evaluations?

AUTHORS: We have clarified in the text that we were not asking parents to give their own judgments of their child’s typical versus atypical developmental history; rather, we were asking them to report on any previous diagnoses provided to them by a doctor or allied health professional. While there are numerous studies of the former type (i.e., does parent report of developmental concerns agree with professional diagnoses), I am aware of no literature of the second type (i.e., do parents accurately respond to questionnaires asking them to report diagnoses assigned to their child by a professional).

3) Materials and Methods: Details about the psychometric fitting results are missing. Authors should provide the exact fitting formula with all customized parameters (if any, e.g., lapses), plots of the average fitted curves and an appropriate goodness of fit measure. The absence of these details reduces the reproducibility of the study and prevents readers from understanding how well the model fitted to data.

AUTHORS: Thank you for this important point. We have expanded the information reported in our online supplementary document with complete results of all regressions. This now includes including the fitting formula for each model and R-squared as a measure of goodness of model fit. 

4) Materials and Methods: Is visual inspection a reliable method to determine whether data are normally distributed or not? Author might provide appropriate tests to justify their point. 

AUTHORS: We have added a statement reporting that we conducted Shapiro-Wilk tests of normality for each measure (identification, category goodness judgment) and modality group (online, in-person) and that the hypothesis of normal distribution was rejected in all cases except the category goodness judgment measure in the online modality. While we have not added the specific results of each test to our manuscript in the interest of brevity, the tests and corresponding q-q plots have been added to the code released along with our manuscript. 

Furthermore, while non-parametric tests are used when comparing means across the online and in-person modalities, linear models are used to evaluate possible effects of different predictors on the dependent variable. A key assumption to implement linear models is normality of model’s residuals. Authors should provide evidence about the normality of residuals to justify the use of linear models. If such condition is not met, they should implement the appropriate analysis method.

AUTHORS: We have added qq-plots and tests of normality of residuals for the linear models to the supplementary materials with complete model results. The hypothesis that the residuals are normally distributed is in fact rejected in each case. However, current statistical thinking (e.g., Schmidt & Finan, 2018) suggests that this is not an issue when (a) the sample size is sufficiently large [greater than 10 observations per variable], (b) the focus is on interpreting coefficients rather than making predictions. Furthermore, Schmidt & Finan (2018) suggest that methods to transform outcomes to satisfy the assumption of normality of residuals can bias model estimates and are not recommended. Because our models meet criteria (a) and (b) stated above, we feel comfortable in our decision to use linear regression models in spite of the observed deviations from normality of residuals. This reasoning is now spelled out in the online supplementary document alongside the reporting of model diagnostics.

5) Discussion: Importantly, author claim that data collected online are no-less valid than data collected in person (lines 541, 542). They back-up this claim by mentioning that identification boundary and category goodness are correlated in the online study but not the in person one (lines 542-545). It is unclear how this finding would justify their claim nor if this specific pattern in the correlation analysis is the only indicator of the validity of data collected online. Relatedly, it is unclear what do the authors exactly mean with the term “validity”. 

AUTHORS: Thank you for these important reflections. To us, the finding that scores on two tasks intended to assess the same construct (in this case, perception of /r/) is a reassuring indicator that our tasks are functioning as intended. However, it is true that we did not carry out an assessment of concurrent validity in the technical sense. Therefore, we have removed language that describes our findings in terms of validity. Our statement under “Conclusions and clinical implications” now reads as follows: “Although the present study did identify areas of difference between measures of child speech perception collected online and in-person, our results suggest the two modalities yield broadly comparable results. We found that two distinct measures of auditory-perceptual acuity for /ɹ/ were significantly correlated in the online data, and a test of the difference in the strength of association across the two contexts failed to reject the null hypothesis of no difference between modalities. Unexpectedly, the association between the two measures was not significant in the in-person data, but this is potentially attributable to a ceiling effect that was more pronounced in the in-person setting.” The final sentence of the paper now reads “In the interim, the present findings provide a caution that online and in-person data cannot automatically be treated as identical, but they also suggest that data collected online are broadly comparable to measures obtained in the lab and have the potential to be more representative.”

They also suggest that this correlational pattern might be due to different numbers of participants reaching ceiling levels between online and in-person modality. Authors should better clarify the link between results and their interpretations in the discussion section or provide additional analyses/metrics to support their claim (e.g., plotting and/or repeating the analyses by excluding participants showing ceiling performance might be a starting point). This appears as being the central point of the paper, therefore it should be centrally and clearly addressed in the discussion section.

AUTHORS: We do not consider it an effective strategy to re-run the analysis without ceiling-level participants because it would drastically reduce our sample size (e.g., removing 39 out of 96 datapoints in the in-person sample) and substantially change the overall shape of the distribution. Instead of conducting such an analysis, we have chosen to temper our wording pertaining to validity and the comparison between samples, as indicated in our response above.

6) Discussion: Results of linear models showed different unexpected patterns concerning several interaction effects of the predictors “Sex”, “Age” and “Modality” which authors mainly attribute to demographic differences in the characteristics of the participants enrolled in the online and the in-person study. It is completely understandable that such differences were not in complete control of researchers considering the COVID-19 pandemic. However, even if statistically significant differences are present in terms of sex and age distributions between online and in-person modalities, considering the sampling bias as the cause of the found interaction effect might be speculative. Authors should attempt at backing-up their claims by providing appropriate evidence from the related literature and/or develop additional analyses to explicitly test if the sampling bias is the source of the unexpected patterns. If possible, authors might try to repeat the analyses by randomly subsampling the in-person database to match the characteristics of the online group in terms of sex and age and verify whether the interaction effects hold. Possible sex differences in speech perception are indeed understudied, thus providing stronger evidence about such patterns might be of relevance to the field.

AUTHORS: Thank you for raising these points; your suggestions helped us refine our analyses and better understand our results. Regarding the decision to include covariates of age and sex in our analyses, Reviewer 2 raised a similar question; please see our response to that reviewer below, where we expand in detail on our theoretical motivation for including these measures in our models. In response to your suggestions for our statistical analyses, we repeated our regressions with weighting by sex and age so that the two groups more closely resemble one another on these characteristics. This change created only minor shifts to our coefficients, which we repeat below for the reviewer’s consideration. We have added a statement to this effect in the paper on p. 26 in the document with tracked changes. The results of the reweighted analysis are not included in the main text, but they can be found in the code released along with our manuscript. In light of the finding that reweighting the model did not change our findings, we have altered any language suggesting that differential patterning of sex in the online versus in-person data could be an “artifact” of sampling differences across modalities (see edits on p. 23, 26, and 28 of the document with tracked changes).

Minor Issues

7) Lines 42-44: The syntax of the sentence is a bit unclear, please rephrase to clarify. 

AUTHORS: Rephrased to read “In particular, this line of research aims to understand the role of speech perception in speech deviations that persist through late childhood or adolescence.”

Additionally, authors should clarify the meaning of “speech deviation” to facilitate the understanding of non-expert readers.

AUTHORS: Added “The speech output of individuals with SSD is characterized by deviations (substitutions, disortions, omissions, and/or additions) that exceed developmental expectations and can negatively impact speech intelligibility.”

8) Line 57: Authors should provide an exemplary English word embedding the /ɹ/ sound as they do later in the manuscript.

AUTHORS: Added “(as in words such as “red” and “deer”)”

9) Lines 61-62: Please clarify the meaning of “the goals driving speech are dynamic regions in auditory-acoustic and somatosensory space”.

AUTHORS: Rephrased to read “According to current neurolinguistic models of speech-motor control such as DIVA [7], HSFC [8], and FACTS [9], humans learn to speak by identifying the auditory-acoustic characteristics associated with a speech sound, then exploring the mapping between movements of the speech structures and perceptual consequences until they find a motor command that maps onto a given perceptual target.”

10) Line 66: Please provide a definition for “auditory and somatosensory targets” to help non-expert readers.

AUTHORS: Rephrased to read “Stored representations of speech sound targets are thought to have both an auditory component (i.e., what should be heard when the sound is produced) and a somatosensory component (i.e., what it should feel like to produce the sound).”

11) Line 139: Typo, please correct ``noisier" with “noisier”.

AUTHORS: Corrected.

12) Line 212: Please provide a reference or an additional explanation backing up the assumption that a stronger correlation between two measures of the same construct in one modality is an index of the robustness of the assessment in that modality.

AUTHORS: This assertion has been deleted. The text now reads “The rationale for this final analysis is that different measures of the same construct (in this case, perception of /ɹ/) are expected to be positively associated, and the degree of association is predicted to be comparable across modalities.”

13) Line 290: Please fix the citation style of (Ortiz, 2018).

AUTHORS: Corrected, thank you.

14) Line 314-315: Typo, please correct ``choose whether the /ɹ/ sound is right or wrong" with “choose whether the /ɹ/ sound is right or wrong”.

AUTHORS: Corrected.

15) Line 332-334: To facilitate the understanding of the rationale of statistical analyses please do not refer to the “first” and “second” research aim but explicitly report again the aims in words.

AUTHORS: Replaced with “To compare average performance across modalities” and “To examine whether the effect of modality differed based on participant characteristics.”

16) Line 468: Please be more specific about the result instead of using “higher performance”.

AUTHORS: Updated to read “Differences in equipment across modalities could potentially account for our finding that in-person participants tended to exhibit narrower boundary widths (suggestive of higher auditory acuity) on the identification task.”

17) Results: While authors provide numerical values of beta coefficients when reporting the results of analyses on the linear models (which are usually self-explanatory), also reporting Mean and Std.Dev/Std.Err. (or other appropriate summary descriptive statistics of choice) of the dependent variables might help readers into mentally representing differences between two/more factor levels.

AUTHORS: We have added plots of estimated marginal means for each regression to the online supplementary materials and noted their inclusion on pages 18-19. 

18) Materials and Methods: In the instruction of the category judgement task authors report that participants saw the “choose whether the /ɹ/ sound is right or wrong” sentence on the screen. Were participants already familiar with the meaning of the “/ɹ/” grapheme or were they given additional explanation about IPA notation? Providing details on instructions, especially in these kinds of tasks developed for children and adolescents might help other researchers in developing efficient instructions

AUTHORS: Thank you for catching our error – we did not use the IPA symbol with our child participants. The text has been updated to read “They received the following instructions: “Listen to the word containing /r/. After hearing a word, choose whether the /r/ sound is right or wrong.” No training was provided, allowing participants to use their own standard for “right” or “wrong” production.”

19) Materials and Methods: Were parents present during the assessments in online/in-person modality? Please specify.

AUTHORS: Parents were not present during assessments in either modality. This has been added at the end of the Stimuli and Protocol section.

20) Factor naming is inconsistent throughout the manuscript. Authors sometime use the term “modality” to refer to the modality in which the experiment was carried out (in-person or online), but some other times they use “condition” (line 373, 387,498) or “group” (line 420, 465). Please use consistent factor and levels naming.

AUTHORS: Thank you for pointing this out. We have changed “condition” references to “modality” and have altered references to “group” to read “modality group.”

21) Figures: Authors provided density plots to represent the found effects in their statistical analyses but they might not be the most appropriate representations, especially when interaction effects are present. The navigation of the results might be facilitated by adding additional figures employing e.g., boxplots/violin plots for categorical variables (e.g., effect of sex) and e.g., points and lines plots when representing the effect of a continuous variable (e.g., age).

AUTHORS: Thank you for this suggestion. We have replaced figure 2 with a violin plot that allows the reader to more easily visualize the differences in median across modality and sex. We have also added a plot representing the interaction between age and modality. As an added note, the plots have been regenerated using colors that differ in luminance, so differences in color remain visible if printed in grayscale.

Reviewer #2: I’d like to thank the authors for their interesting and well executed study. The methodology is rigorous, and the authors have written with great clarity and precision, which greatly aids in the potential reproduction of the current study. Additionally, I thank the authors for sharing their data and code, which has been presented with detailed metadata and appropriate comments throughout the code to guide the reader through the analysis.

AUTHORS: We appreciate your constructive feedback!

Understanding the impact of research setting (lab vs. online) is of great importance, particularly in light of the COVID-19 pandemic and the increasing shift toward collecting perceptual data remotely. I thank the authors for their valuable contributions in this area.

My primary concern/critique relates to the second research question, specifically regarding the examination of the interaction between age and sex. In its current form, the introduction lacks sufficient justification or rationale for considering the listener sex in the analysis. Moreover, in the discussion section, the authors acknowledge limited evidence supporting the idea that biological sex might influence perceptual acuity. To strengthen the study's introduction and rationale, I’d recommend incorporating this discussion point into the introduction.

As it stands, the investigation into sex differences appears to rely solely on the presence of observed differences in the results. However, without a well-defined rationale for exploring these sex differences, it becomes challenging to discern whether these disparities are indicative of a genuine sex-effect or merely a consequence of dividing the two groups into further subgroups. Clarifying the motivation behind examining sex differences is crucial to the overall validity and interpretation of the study's findings.

AUTHORS: Thank you for this important observation. We did have an a priori motivation to examine listener sex as well as age based on previous research findings. We now state this motivation explicitly in the introduction section, as follows: “There is also evidence that the association between perceptual acuity and treatment response could be moderated by factors such as age and sex. For instance, Cialdella et al. [3] found a significant association between auditory-perceptual acuity and treatment response in female but not male participants in a retrospective analysis of 59 participants aged 9-15 who received treatment for RSSD.” We also tried to clarify the role of previous research in motivating the analysis by age and sex when stating the purpose of the study (“Second, we investigated whether there were any interactions between modality and participant age or sex, since previous research suggests that impacts of modality could differ over maturation or across speaker subgroups [55,58]”) and when stating our plan for analysis (“In light of these differences between the modality groups, as well as previous research suggesting that factors such as age and sex may matter for perceptual measurement [3,55], below we will examine models of perceptual performance that control for age and sex.”) We have also reframed our statement in the Discussion to avoid incongruity: “We chose to examine sex and age as predictors based on a small number of previous findings suggesting that biological sex can influence perceptual acuity at a basic level [69] or can interact with perceptual acuity in predicting treartment response [3].” We hope this is sufficient to communicate to the reader that our decision to analyze modality in interaction with age and sex was motivated not only by the presence of heterogeneity in our data around those variables, but also by previous literature providing a theoretical motivation for examining them.

For clarity, can the authors give an example of what type of listener response would result in a boundary of 0.0?

AUTHORS: Under the description of the identification task, we have added “A boundary width of zero means that no more than one continuum step was classified with any degree of inconsistency,” and in the reporting of ceiling-level results, we have added a parenthetical “a boundary width of 0.0 (which means that 0 or 1 stimuli were classified with less than 100% consistency).”

The authors stat that the calculated measures were examined for outliers using a criteria of > 3 MAD and reported no outliers were observed. However, there does seem to be some influential bivariate outliers for the correlation analysis (as observed in Figure 4). Could the authors comment on these observations and their decision not to exclude these cases? Did the presence of these outliers inform the decision to use Spearman’s rho?

AUTHORS: We added a footnote on this point: “Some datapoints in Figure 4 present visually as outliers, although they were not flagged as such on the basis of median absolute deviation from the group median. However, the influence of these datapoints is limited by our decision to use Spearman’s rho as our measure of association.”

Minor edits:

There appears to be a word missing in the first paragraph of the introduction, line 41.

AUTHORS: Fixed, thank you! 

In line 364 of the results, when discussing the study-domain effects for male listeners, I’d recommend just saying that the difference between online and in-person modalities was not significant for male participants of average age. This revised sentence would parallel the structure of the previous sentence discussion the study-domain effects for the female listeners.

AUTHORS: Thank you for the suggestion. The text now reads “However, the difference between online and in-person modalities was not significant for male participants of average age.”

---

## [Decision Letter · Decision Letter 1]

8 Jan 2024

Comparing online versus laboratory measures of speech perception in older children and adolescents

PONE-D-23-23646R1

Dear Dr. McAllister,

We’re pleased to inform you that your manuscript has been judged scientifically suitable for publication and will be formally accepted for publication once it meets all outstanding technical requirements.

Kind regards,

Simone Sulpizio

Academic Editor

PLOS ONE

Additional Editor Comments (optional):

Reviewers' comments:

Reviewer's Responses to Questions

**Comments to the Author**

1. If the authors have adequately addressed your comments raised in a previous round of review and you feel that this manuscript is now acceptable for publication, you may indicate that here to bypass the “Comments to the Author” section, enter your conflict of interest statement in the “Confidential to Editor” section, and submit your "Accept" recommendation.

Reviewer #1: All comments have been addressed

Reviewer #2: All comments have been addressed

2. Is the manuscript technically sound, and do the data support the conclusions?

Reviewer #1: Yes

Reviewer #2: Yes

3. Has the statistical analysis been performed appropriately and rigorously? 

Reviewer #1: Yes

Reviewer #2: Yes

4. Have the authors made all data underlying the findings in their manuscript fully available?

Reviewer #1: (No Response)

Reviewer #2: Yes

5. Is the manuscript presented in an intelligible fashion and written in standard English?

Reviewer #1: Yes

Reviewer #2: Yes

6. Review Comments to the Author

Reviewer #1: (No Response)

Reviewer #2: I’d like to thank the authors for their revision. I believe the revised manuscript sufficiently addresses my original critiques, as well as the critiques made by the other reviewer.

Specifically, the addition of the previous literature’s findings of the significant impact of factors like sex and age has addressed my previous critique about the authors lacking a rationale for including these variables in their analyses.

Further, the expanded explanation of the interpretation of the identification task (0-1) greatly helps the interpretation of this measure and the findings.

Regarding the critique made by the other reviewer regarding the assumption of normality of residuals in the linear models, I support the authors’ response, citing that the violation of this assumption is not an issue when the sample size is appropriate.

Finally, I appreciate the other reviewer’s critique about the use of “validity,” given that the type of validity was not specified and a test of validity was not conducted explicitly. However, I believe that the authors’ modifications to the manuscript adequately address these concerns.

I believe the revised manuscript is a strong study.

7. PLOS authors have the option to publish the peer review history of their article (what does this mean?). If published, this will include your full peer review and any attached files.

Reviewer #1: **Yes: **Giuseppe Di Dona

Reviewer #2: No

---

## [Editor Report · Acceptance letter]

30 Jan 2024

PONE-D-23-23646R1 

PLOS ONE

Dear Dr. McAllister, 

I'm pleased to inform you that your manuscript has been deemed suitable for publication in PLOS ONE. Congratulations! Your manuscript is now being handed over to our production team.

Kind regards, 

on behalf of

Dr. Simone Sulpizio 

Academic Editor

PLOS ONE